# Correlates of Zero-Dose Status among Children Aged 12–23 Months in the Luambo Health District, Democratic Republic of Congo: A Matched Case–Control Study

**DOI:** 10.3390/vaccines12070700

**Published:** 2024-06-21

**Authors:** Esperent Ntambue Malu, Alain Nzanzu Magazani, Jean Bosco Kasonga, Adèle Mudipanu, Michel Kabamba Nzaji, Daniel Katuashi Ishoso, Dalau Mukadi Nkamba

**Affiliations:** 1Kasaï Central Provincial Health Division, Ministry of Public Health, Hygiene and Prevention, Kananga 05101, Democratic Republic of the Congo; 2African Field Epidemiology Network (AFENET), Kinshasa 01207, Democratic Republic of the Congo; alain.magazani@gmail.com; 3Kinshasa School of Public Health, University of Kinshasa, Kinshasa 01302, Democratic Republic of the Congo; jbkas24@gmail.com (J.B.K.); dishosok@gmail.com (D.K.I.); dalau.nkamba@unikin.ac.cd (D.M.N.); 4United Nations International Children’s Emergency Fund (UNICEF) Country Office, Kinshasa 01204, Democratic Republic of the Congo; amudipanu@unicef.org; 5Expanded Program of Immunization, Kinshasa 01208, Democratic Republic of the Congo; michelnzaji@yahoo.fr; 6World Health Organization (WHO) Country Office, Kinshasa 01205, Democratic Republic of the Congo

**Keywords:** Luambo health district, Democratic Republic of the Congo, immunization, zero-dose, correlates

## Abstract

(1) Background: “Zero-dose” (ZD) refers to a child who has not received any doses of the pentavalent (diphtheria–tetanus–pertussis–*Haemophilus influenzae* type b (Hib)–hepatitis B) vaccine. ZD children are vulnerable to vaccine-preventable diseases (VPDs). Luambo health district (HD) is one of 26 HDs in Kasai Central Province in Democratic Republic of the Congo and had the largest number of ZD children in 2021. This study was conducted to identify factors associated with ZD status among children in Luambo HD. (2) Methods: We conducted a mixed-methods study of children aged 12–23 months in Luambo HD. (3) Results: A total of 445 children aged 12–23 months were included in the study, including 89 cases and 356 controls. Children who were born in Angola (AOR = 3.2; 95% CI = 1.1 to 9.8; *p* = 0.046), born at home (AOR = 5.2; 95% CI = 2.1 to 12.5; *p* < 0.001), whose mothers did not receive antenatal care (AOR = 4.4; 95% CI = 1.2 to 16.3; *p* = 0.023), or did not know any vaccine preventable disease (AOR = 13.3; 95% CI = 4.6 to 38.4; *p* < 0.001) were more likely to be ZD than their counterparts. In addition, perceptions of children’s parents influenced child immunization. (4) Conclusions: Factors associated with being a ZD child suggest inequalities in vaccination that need to be addressed through appropriate interventions. Maternal and child health services need to be strengthened while also targeting children’s fathers. This will make it possible to considerably reduce the proportion of ZD and undervaccinated children and effectively fight against VPDs.

## 1. Introduction

Routine childhood immunization is a proven strategy for controlling and eliminating life-threatening infectious diseases [1]. According to the World Health Organization (WHO), vaccination prevents around four million deaths each year. “Zero-dose” (ZD) refers to a child who has not received a single dose of the pentavalent (diphtheria–tetanus–pertussis–*Haemophilus influenzae* type b (Hib)–hepatitis B) vaccine during the routine national immunization schedule [2]. ZD children are vulnerable to vaccine-preventable diseases (VPDs) [1,2].

From 2019–2020, the number of ZD children varied by WHO region. This number remained stable in the European region at 0.3 million but increased in the African region (from 7.1 million to 7.7 million), American region (from 1.6 million to 1.7 million), Eastern Mediterranean region (1.8 million to 2.3 million), South-East Asian region (2.0 million to 4.1 million), and Western Pacific region (0.9 million to 1.0 million). In 2020, low- and middle-income countries accounted for the highest proportion of ZD children (71%; 12.1 million); of these, 4.1 million children (24%) were from countries in the African region and the South-East Asian region [3]. Four and a half million ZD children (26%) lived in low-income countries [2,3].

In 2020, during the COVID-19 pandemic, there was insufficient access to vaccination and other health services. Twenty-three million children worldwide did not receive the vaccines they should have received. Among them, 17.1 million were ZD children and another 5.6 million were only partially vaccinated. Of these 23 million children, more than 60% lived in 10 countries: Angola, Brazil, Ethiopia, India, Indonesia, Mexico, Nigeria, Pakistan, Philippines, and Democratic Republic of the Congo (DRC) [1].

According to vaccination coverage surveys carried out in DRC, the proportion of ZD children among children aged 12 to 23 months increased from 9.2% to 12.7% between 2014 and 2021 [4,5,6,7]. The proportion of ZD children among children aged 12 to 23 months in Kasaï Central Province was 12.6% in 2021 [7]. In Kasaï Central Province, Luambo health district (HD) had the highest proportion of ZD children (44.5%), despite the implementation of the Mashako plan in 2020, an emergency plan aimed at relaunching routine vaccination activities to increase vaccination coverage in DRC [7,8]. Immunization coverage in DRC remains well below the global target of 90% according to national surveys and WHO/UNICEF estimates [9].

The purpose of this study was to investigate factors associated with ZD status among children aged 12 to 23 months in Luambo HD.

## 2. Materials and Methods

### 2.1. Study Setting

The Luambo HD is one of 26 HDs in Kasai Central Province. The HD is 285 km (km) from Kananga, the capital city of the province, with an estimated population of 349,793 inhabitants over an area of approximately 4320 km^2^, yielding a density of 81/km^2^. Luambo HD borders Angola and hosts a transient population from Angola [10]. Luambo HD has 21 health areas (HAs), 305 villages, and 450 community animation cells (CACs) [10]. HAs organize routine immunization services and CACs are groups of community health workers that support community participation in the effective management of health services. 

### 2.2. Study Design

We conducted a mixed-methods study with a matched case–control study and a qualitative descriptive component among children aged 12–23 months in Luambo HD.

### 2.3. Study Population

Children aged 12 to 23 months residing in study HAs at the time of data collection whose mothers or caregivers consented to participate in the study were included in this study. A ZD child aged 12 to 23 months was considered a case. We considered the following controls:
Any child aged 12–23 months living in the study HAs who received at least one dose of pentavalent before their first birthday but who did not complete the vaccination schedule recommended by the National Expanded Program of Immunization (EPI).Any child aged 12–23 months fully vaccinated before the age of 12 months according to the EPI recommendation and living in the study HAs at the time of data collection [11].

### 2.4. Sampling

We estimated the desired sample size of children aged 12–23 months in the study using Stata software, version 15. We used mother missing antenatal care (ANC) during pregnancy as a correlate of ZD status in children aged 12–23 months with odds ratio (OR) of 2.13 [12], correlation coefficient of 0.2 between matched cases and controls [13], noninformative proportion (50%) of controls for whom mothers missed ANC during pregnancy, power of 80%, ratio of 1 case to 4 controls, and nonresponse rate of 10%. The required sample size was thus 445 (including 89 cases and 356 controls).

We carried out a three-stage cluster sampling process. During the first stage, we randomly selected 8 HAs among the 21 HAs in Luambo HD using a random number generator. The number of cases and controls was distributed equally among all HAs, therefore, 11 cases and 44 controls were surveyed per HA, except in Kasombo Bishi HA where 12 cases and 48 controls were surveyed. During the second stage, we selected 4 CACs from each HA, for a total of 32 CACs. During the third stage, cases were selected by systematic sampling in each selected CAC. In each CAC, with the help of community health workers, we drew up a list of all children aged 12–23 months. For each child, the record indicated ZD status, sex, and age in months. We numbered all ZD children from 1 to Ni, with Ni representing the total number of ZD children in the i^th^ CAC. We calculated the sampling specific step for each CAC, accounting for the number of ZD children by dividing them by the number of ZD children required for the CAC. There were 3 cases per CAC in 3 CACs and 2 cases in 1 CAC. To obtain the CAC for 2 children, a random draw was carried out on the 4 CACs. To select the first ZD child to be surveyed, we randomly selected a number between 1 and the integer part of the sampling increment. Others ZD children were selected by sequentially applying the sampling step. For each ZD child selected, we selected 4 same-sex controls adjacent to the case on the child record: 2 fully vaccinated and 2 undervaccinated children. In the qualitative component of the study, we selected a group of mothers of children aged 12–23 months. Three focus groups of 8 mothers of children aged 12–23 months were conducted: one with mothers of fully vaccinated children, one with mothers of ZD children, and one with mothers of undervaccinated children.

### 2.5. Data Collection

In the quantitative study, data were collected through face-to-face interviews with mothers or caregivers, using a structured questionnaire on paper. Interviews were typically conducted in the morning and evening, the time when respondents were most likely to be at home.

In the qualitative study, we collected data using a dictaphone and a previously developed interview guide. Interviews were conducted in Tshiluba, the local dialect of Luambo General Referral Hospital in Luambo HA. Interviews lasted between 40 and 60 min. On days when qualitative data collection took place, it occurred after all quantitative data collection was conducted for the day. Interviews took place on a taboo day, a day when parents are forbidden to go to the fields according to custom. All participants came from Luambo HA. Interviews were conducted by the principal investigator, using the interview guide to collect data about perceptions of child immunization. The principal investigator was assisted by a note-taker who was a supervising nurse from the Luambo HD. All interviews were recorded on audiotape using a dictaphone and then uploaded to a secure box folder. All participants were given a unique identifier so that the transcriptions could be deidentified. The interviews were transcribed into French for analysis.

### 2.6. Variables

The outcome of interest was ZD status. The explanatory variables used were sociodemographic characteristics of the head of household (e.g., age, level of education, profession, and religion), characteristics of the mother/caregiver (e.g., her relationship with the child, age, marital status, level of education, occupation, religion, reception of ANC during pregnancy), number of children in the household, knowledge of VPD, and characteristics of the child (e.g., gender, age, upper arm circumference, place of birth, country of birth, and sibling rank).

### 2.7. Data Analysis

We manually entered data into Epidata 3.1 software and exported them to Stata 15 software for analysis. We summarized categorical variables as proportions and summarized quantitative variables using median with interquartile range (IQR) as they were not normally distributed. We used the Mann–Whitney test to compare medians between cases and controls. We used the chi-square test of homogeneity to compare categorical variables between cases and controls. We used conditional logistic regression to identify factors associated with ZD status. All variables with a *p*-value less than or equal to 0.20 in simple regression were candidates for multiple regression. We checked multicollinearity among the independent variables using the variance inflation factor (VIF). A VIF equal to or greater than 10 was indicative of multicollinearity. We found no collinearity between independent variables. Study results are presented as crude odds ratio (OR) and adjusted odds ratio (AOR) with 95% confidence intervals (95% CI). The level of statistical significance was set at 0.05.

Records from focus groups were listened to several times and transcribed onto a summary sheet. The written transcriptions were entered into the qualitative data management software ATLAS.ti version 7 (2013) (Scientific Software Development, Berlin, Germany) and coded according to a code dictionary based on the themes included in the transcription guides. We first performed thematic analysis for mothers of fully vaccinated children, mothers of undervaccinated children, and mothers of ZD children. Next, we developed matrices to facilitate comparisons between transcripts and to maintain the context of the data. Finally, the data were summarized and interpreted. We selected direct quotes representative of the participants’ opinions to illustrate the results without providing the identity of the participants.

### 2.8. Ethical Approval and Consent to Participate

The ethics committee of Kinshasa School of Public Health approved this study before data collection (approval number: ESP/CE/16/2023). Authorization was also granted by health and political-administrative authorities. Before beginning interviews, written informed consent was obtained from study participants. The research team provided each respondent with information on the nature of the study, its objectives, risks and benefits involved, the freedom to participate or not without any prejudice, confidentiality, and contact details of the person in charge of the study. Confidentiality was respected by anonymizing the dataset.

## 3. Results

### 3.1. Quantitative Results

#### 3.1.1. Sample Characteristics

The study included a total of 445 children aged 12–23 months, including 89 cases and 356 sex-matched controls, with a ratio of 1 case to 4 controls. Table 1 shows the characteristics of household heads. The median age of heads of household was 34 years (interquartile range (IQR) = 27–41) for cases and 35 years (IQR = 28–41) for controls. More than eight out of ten heads of household were male in both cases and controls. Approximately four of ten respondents had never been to school among cases, compared with two of ten among controls (42.7% vs. 20.8%).

Table 2 shows the characteristics of the mothers/caregivers. The median age of mothers/caregivers was 28 years (IQR = 23–33) for cases and 27 years (IQR = 23–32) for controls. Among respondents, around 98% were mothers of the children in cases compared to 97% in controls. Approximately one in five case respondents lived in a union. More than six out of ten respondents had never been to school among cases, whereas only four out of ten had done so among controls (66.3% vs. 42.7%). Approximately nine out of ten respondents had a gainful occupation in both cases and controls. Approximately only six out of every hundred mothers in cases had made adequate ANC visits (≥4) during their pregnancy among cases while thirty-eight out of every hundred mothers among controls had not made any ANC visits during pregnancy. More mothers of controls were aware of at least one VPD than mothers of cases (95% vs. 45.2%).

Table 3 shows the characteristics of the children aged 12–23 months in our study. The median age was 17 (20–14) months in ZD children and 17 (19–14) months in controls. In addition, the median upper arm circumference (MUAC) was 128 cm (IQR = 120–132) in cases and 132 cm (IQR = 127–140) in controls. The proportion of children born in Angola varied from 6.7% in controls to 40.4% in the ZD group. About 58% of ZD children and 7% of control children were born at home (58.4% vs. 6.7%). None of the cases had a vaccination card.

#### 3.1.2. Factors Associated with ZD Status in Children Aged 12 to 23 Months in the Luambo HD

In multivariable regression analysis, being born in Angola (AOR = 3.2; 95% CI = 1.1 to 9.8; *p* = 0.046), being born at home (AOR = 5.2; 95% CI= 2.1 to 12.5; *p* < 0.001), lack of awareness about VPDs by the mother (AOR = 13.3; 95% CI = 4.6 to 38.4; *p* <0.001), and lack of ANC visits by the mother during pregnancy with the indexed child (AOR = 4.4; 95% CI = 1.2 to 16.3; *p* = 0.023) significantly increased the odds of being ZD among children aged 12 to 23 months. A summary of these findings is in Table 4.

### 3.2. Qualitative Results

Group discussions revealed that mothers’ perceptions influence children’s vaccination. According to mothers of fully immunized children, the majority of mothers vaccinate their children but they do not follow the vaccination schedule. On the other hand, a few of the interviewed mothers confirmed that some mothers who refuse to vaccinate their children do so because of fear of fever and incessant crying. A mother declared: “*No, they all vaccinate, but it is just that many stop vaccinating their children*”. Another mother put it another way: “*Yes, many refuse because of the children’s crying and the pain caused by vaccines. They say vaccines make children sick*”.

Fathers also play an important role in the vaccination of their children. There are husbands who, because of their negative perception of vaccination, refuse to allow their wives to visit the health center for ANC and for vaccination of their children. One mother said: “*They tell you that since the time of our grandparents and great-grandparents, they did not get vaccinated, were they not growing up well? They grew up without problems*”. Another mother added: “*Others say that they have children so that the nurses can start taking money from their children, while they don’t benefit from anything. Nurses and community health workers are paid when their children are vaccinated while the parents are not. Therefore, to avoid enriching nurses on their children’s backs, it is better not to bring children for vaccination. Therefore, we work in vain to make the nurses and community health workers rich. We only remember the community health workers and the nurses without remembering those who make these children. You who waste this money should vaccinate your children*”.

For mothers of ZD and undervaccinated children, the following reasons were given for not following the vaccination schedule: fear of side effects (crying and pain at the injection site), the child’s mother’s illness, lack of vaccines at the health center, AEFI (swelling in the right thigh, fever), expensive care after vaccination, distance between home and the health center, absence of the child’s father, and lack of moral support. A mother of an undervaccinated child declared that: “*Because every time we vaccinated him, he got sick until he was taken to the health center again for treatment. And every time I took him to the health center, they demanded to pay for the treatment. So, I had to give up, because it was a lot of work*”. A mother of a ZD child added: “*I was afraid because of my child who comes immediately before this one. He had fallen seriously ill after receiving the vaccines. We left with his eldest and after receiving the vaccines, he suffered a lot. Therefore, we were afraid that this one would also fall ill like his predecessor*”.

When asked what would happen to a child who is not vaccinated, most mothers of ZD children believe that neither humans nor vaccines have any power over a child’s life. It is God who decides on the lives of humans because illness does not choose children to catch. Whether he is vaccinated or not, he will catch the disease. All children get sick. As one mother put it: “*Everything depends on God. Whether the child is vaccinated or not, if God decides he’s going to die, he’ll die even if he’s vaccinated. Even if he is not vaccinated, nothing can happen to him without God’s will*”.

Regarding the importance of childhood vaccination, the majority of ZD mothers noted that they were unaware of the importance because they had not heard about the necessity of vaccinating children during their pregnancy. Particularly, mothers that lived in Angola said they did not receive information about the importance of childhood vaccination. A minority recognized the importance of vaccination. One mother declared: “*I too don’t know the importance of vaccinating children. In Angola where I was, I had never heard of it*”. Another mother added: “*I never knew the importance of vaccination. Moreover, I’m not interested in vaccinating children*”. 

With regard to the perception of vaccination by those around them, mothers of children noted that people’s opinions are divided. A mother of an undervaccinated child said: “*There are those who appreciate vaccinating children. Sometimes, you can be in a state of lethargy and you see someone who comes to make you aware to tell you: “but we shouted that there is vaccination of children today but why are you dragging to bring the child to the vaccination? Because vaccines are effective and help children a lot*”. Another added in these terms: “*When they see that the child is sick after having been vaccinated, they tell you: “that we told you that vaccines cause illnesses in children but you did not bring your child. I cannot bring my child for vaccination. Now, you will have to go and pay the costs of care at the health center*”. A mother of a ZD child commented: “*In my neighborhood, a lot of people say it’s not a good idea to have children vaccinated. Vaccines cause many illnesses in children. Others also agreed that is good to send children for vaccination. Opinions differ between people*”.

## 4. Discussion

The present study was carried out to identify correlates of ZD status of children aged 12–23 months in the Luambo HD in Kasai Central Province. This study showed that accessibility of adequate ANC, lack of awareness about VPD, and being born in Angola or at home were correlated with ZD status among children aged 12 to 23 months. The study observed that mothers who had not used ANC services were more likely to have ZD children compared to mothers who had completed recommended ANC visits (≥4). These results are in line with those found in previous studies which showed a statistically significant association between the number of ANC visits attended and childhood immunization [14,15,16,17]. ANCs are an opportunity for future mothers to receive advice from healthcare providers on consequences of pregnancy, childbirth, and care for newborns, including vaccination. Mothers who do not attend ANC visits are deprived of this advice, which may explain why they are more likely to have a ZD child.

Mothers unaware of any VPD were more likely to not vaccinate their children. This result is similar to that of Udessa G et al. in Ethiopia [18]. This association can be explained by the fact that childhood vaccination is a practice and one determinant of good practice is knowledge [18]. Lack of knowledge about vaccination and VPD can decrease or inhibit the practice of childhood vaccination and decrease perceived susceptibility to VPD, therefore putting children at risk of VPD.

The study also showed that being born at home increases the likelihood of a child to be ZD compared with being born in a healthcare facility. Similar results were published in previous studies in DRC [19], East Africa [20], Guinea [14], Cameroon [21], India [22], and Turkey [23]. This can be explained by the fact that mothers who give birth at home are mainly those who do not attend ANC visits during their pregnancies. In addition, children born at home do not receive a vaccination card, which is a tool for planning future appointments. In addition to having a safe delivery by health professionals, giving birth in a healthcare establishment provides the additional advantage of postdelivery follow-up and access to vaccination-related information provided by health professionals. During the childbirth visit, healthcare providers can provide maternal education on vaccination and other maternal and child healthcare, which could increase mothers’ practice of vaccinating their children [19]. This finding highlights the need to increase awareness among pregnant women about the importance of ANC and institutional delivery and the impact on child immunization. Nevertheless, as long as some women continue to give birth at home, it will be necessary to pay special attention to these children and administer the necessary vaccines. In these communities, a context-specific approach, such as using mobile clinics and working with traditional birth attendants to identify and vaccinate children at birth, will significantly improve immunization coverage.

The study also showed that children born in Angola were more likely to be ZD. Although the reason for this finding was not directly collected, we speculate that most mothers who travel to Angola give birth at home because many live in mining camps and do not have access to maternal and child care. Vaccination services in both countries (Angola and DRC) should therefore tailor intervention to reach children in mining camps via mobile or advanced vaccination sessions, as the place of residence of the mother plays a large role in the child’s vaccination [24].

Unlike other studies that have demonstrated a significant association between maternal education and child vaccination status [17,20,25,26,27], our study found no significant association between maternal education and the ZD status of children. This could be due to the fact that although study participants attended school, they retained negative beliefs and perceptions about childhood vaccination, as illustrated by the focus group results.

The study also showed that the interviewed mother’s perceptions of child vaccination varied. Some mothers had a good perception and believed that through vaccination, children are immunized against VPD and will have fewer health problems that can weaken their growth. However, mothers of ZD children believed that vaccines do not help children and instead put families at risk by causing fevers and other illnesses that deprive families and lead to paying for medical care. Some of the interviewed mothers believed that neither men nor vaccines have any power over the life of a child. It is God who decides on the lives of humans because illness does not choose children to catch. Whether the individual is vaccinated or not, catching the disease or not depends on God. Negative perceptions of childhood vaccination among mothers and fathers appeared to be linked to habits and customs and lead them to stop vaccinating their children or not to vaccinate them at all. This negative perception is also linked to a deficit of communication between vaccination service providers and mothers and/or babysitters. Health professionals may not communicate sufficiently with mothers, especially regarding AEFI. This insufficiency leads parents to stop vaccinating their children. Given that some mothers do not attend ANC visits, educational talks focused on vaccination should be organized in the community with the mothers and fathers of children to help them understand the merits of vaccination and thus improve coverage vaccination in Luambo HD.

### Strengths and Limitations

The greatest strength of our study is the use of mixed methods (quantitative and qualitative). However, the case–control design does not make it possible to establish a causal relationship. Additionally, this study did not assess supply-side factors such as type of nearby health facility, type of health professional providing vaccination services, availability of vaccines, and supply shortage periods in the HD. These supply factors likely influence child immunization. Finally, the study did not collect data on reasons for birth at home.

## 5. Conclusions and Recommendations

Missing ANC during pregnancy and a lack of knowledge about childhood VPD increased the odds of ZD status among children in Luambo HD. This study highlights the need to develop context-specific approaches to vaccinating children through the creation of health posts and mobile clinics and collaboration with traditional birth attendants. Furthermore, interventions to increase vaccination uptake should target not only mothers but also fathers of young children. Finally, the introduction of a free VPD support package in routine EPI as suggested by some mothers would likely reduce the number of under vaccinated children.

## Figures and Tables

**Table 1 vaccines-12-00700-t001:** Sociodemographic characteristics of household heads.

Characteristic	Case (n_1_ = 89) n (%)	Control (n_2_ = 356) n (%)	*p*
Age (years)			
Median and IQR	34 (41–27)	35(41–28)	0.905
Age group (years)			0.943
17–24	15 (16.9)	62 (17.4)	
25–34	35 (39.3)	133 (37.4)	
35 and over	39 (43.8)	161 (45.2)	
Gender			0.951
Male	73 (82.0)	291 (81.7)	
Feminine	16 (18.0)	65 (18.3)	
Religion			0.005
Catholic	18 (20.2)	138 (38.8)	
Protestant	13 (14.6)	42 (11.8)	
Others	58 (67.2)	176 (49.4)	
Ethnic group			0.813
Lualua	44 (49.4)	171 (48.0)	
Others	45 (50.6)	185 (52.0)	
Level of study			<0.001
Never went to school	38 (42.7)	74 (20.8)	
Primary and more	51 (57.3)	282 (79.2)	
Occupation			0.046
Unpaid profession	18 (20.2)	44 (12.4)	
Remunerative profession	71 (79.8)	312 (87.6)	

n = number of subjects; % = percentage; IQR = interquartile range.

**Table 2 vaccines-12-00700-t002:** Sociodemographic characteristics of mothers/caregivers.

Characteristic	Case (n_1_ = 89) n (%)	Control (n_2_ = 356) n (%)	*p*
Age (years)			
Median and IQR	28 (33–23)	27 (32–23)	0.264
Age group (years)			0.720
17–24	34 (38.2)	152 (42.7)	
25–34	42 (47.2)	159 (44.7)	
35 and over	13 (14.6)	45 (12.6)	
Bond with child			0.673
Mother	87 (97.7)	345 (96.9)	
Babysitter	2 (2.3)	11 (3.1)	
Marital status			0.326
Living alone	9 (10.1)	25 (7.0)	
In union	80 (89.9)	331 (93.0)	
Religion			0.001
Catholic	18 (20.2)	138 (38.8)	
Protestant	13 (14.6)	42 (11.8)	
Others	34 (38.2)	76 (21.3)	
Ethnic group			0.962
Lualua	42 (47.2)	167 (46.9)	
Others	47 (52.8)	189 (53.1)	
Level of study			<0.001
Never went to school	59 (66.3)	152 (42.7)	
Primary and more	30 (33.7)	204 (57.3)	
Occupation			0.007
Unpaid profession	11 (12.4)	27 (7.6)	
Remunerative profession	78 (87.6)	329 (92.4)	
Number of ANC visits (n _1_ = 87 and *n* _2_ = 345)			<0.001
Adequate visits (≥4)	5 (5.8)	132 (38.3)	
Inadequate visits (1–3)	25 (28.7)	147 (42.6)	
No visits	57 (65.5)	66 (19.1)	
Knowledge of VPDs			<0.001
Yes	38 (42.7)	339 (95.2)	
No	51 (57.3)	17 (4.8)	

n = number of subjects; % = percentage; IQR = interquartile range.

**Table 3 vaccines-12-00700-t003:** Characteristics of children aged 12–23 months.

Characteristic	Case (n_1_ = 89) n (%)	Control (n_2_ = 356) n (%)	*p*
Age (months)			
Median and IQR	17 (20–14)	17 (19–14)	0.548
Upper arm circumference (cm)			
Median and IQR	128 (132–120)	132 (140–127)	<0.001
Gender			0.999
Male	43 (48.3)	172 (48.3)	
Feminine	46 (51.7)	184 (51.7)	
Child’s country of birth			<0.001
Angola	36 (40.4)	24 (6.7)	
DRC	53 (59.6)	332 (93.3)	
Place of birth			<0.001
Residence	52 (58.4)	26 (7.3)	
Healthcare establishment	37 (41.6)	330 (92.7)	
Vaccination card			<0.001
Yes	0 (0.0)	285 (80.1)	
No	89 (100.0)	71 (19.9)	
Rank in siblings			0.487
1–2nd	34 (38.2)	122 (34.3)	
3rd and above	55 (61.8)	234 (65.7)	

n = number of subjects; % = percentage; IQR = interquartile range; DRC = Democratic Republic of the Congo.

**Table 4 vaccines-12-00700-t004:** Bivariate and multivariate analysis of factors associated with “zero-dose” status among children aged 12–23 months in Luambo Health District.

Features	Bivariate Analysis			Multivariate Analysis		
	OR	95% CI	*p*	AOR	95% CI	*p*
Age (years) head of household	1	0.9–1.0	0.788			
Gender of head of household						
Male	1	-				
Feminine	0.9	0.5–1.8	0.937			
Religion of head of household						
Catholic	1	-		1	-	
Protestant	3.7	1.4–7.6	0.006	3.8	0.9–16.8	0.074
Others	2.9	1.6–5.6	0.001	2.7	0.8–8.3	0.087
Ethnicity of head of household						
Lualua	1	-				
Others	0.9	0.4–1.9	0.710			
Education level of head of household						
Never went to school	1	-		1	-	
Primary and more	0.3	0.2–0.6	**<0.001**	0.8	0.3–2.1	0.709
Profession of head of household						
Unpaid profession	1	-		1	-	
Remunerative profession	0.5	0.3–1.0	**0.062**	0.8	0.3–2.0	0.589
Age (years) of mothers/guardians	1.0	0.9–1.0	0.434			
Marital status of mothers/guardians						
Living alone	1	-				
In union	0.7	0.3–1.5	0.346			
Education level of mothers/caregivers						
Never went to school	1	-		1	-	
Primary and more	0.3	0.2–0.6	**<0.001**	0.9	0.4–2.2	0.837
Occupation of mothers/caregivers						
Unpaid profession	1			1		
Remunerative profession	0.6	0.3–1.4	**0.020**	0.7	0.2–2.4	0.557
Number of ANC visits						
Adequate visits (≥4)	1			1		
Inadequate visits (1–3)	4.8	1.8–12.7	**0.002**	3.2	0.9–10.8	0.056
No visits	21.5	8.1–57.4	**<0.001**	4.4	1.2–15.4	**0.023**
Knowledge of at least one VPD						
Yes	1	-		1	-	
No	29.9	12.8–69.9	**<0.001**	13.3	4.6–38.4	**<0.001**
Child’s country of birth						
DRC	1	-		1	-	
Angola	11.6	5.7–23.6	**<0.001**	3.2	1.1–9.8	**0.046**
Place of delivery of the child						
Healthcare establishment	1	-		1	-	
Residence	16.3	8.4–31.4	**<0.001**	5.2	2.1–12.5	**<0.001**
Rank in the child’s siblings						
1–2nd	1	-				
3rd and above	0.8	0.5–1.4	0.456			

OR = odds ratio; AOR = adjusted odds ratio; 95% CI = 95% confidence interval.

## Data Availability

The datasets used and/or analyzed during the current study are available from the corresponding author on reasonable request.

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
