# Peer review of "Correlates of Zero-Dose Status among Children Aged 12–23 Months in the Luambo Health District, Democratic Republic of Congo: A Matched Case–Control Study"

_vaccines, 2024, doi:10.3390/vaccines12070700_

Round 1
Reviewer 1 Report
Comments and Suggestions for Authors
Dear Editor-in-Chief,
I have now read the manuscript entitled: “Correlates of zero-dose status among children aged 12-23 months in the Luambo health district, Democratic Republic of Congo: a matched case-control study“ Vaccines-2874014 by Malu EN et al.
Comments: The study aim at investigating the reasons and explanations for weak vaccination frequencies in small children (age 12-23 months) in the Luambo health district in Democratic Republic of Congo during a two-year study period, 2019 and 2020. Performing these types of highly relevant and important studies is highly valuable to identify and discover reasons and explanations to why health care efforts, such as vaccinations, provide suboptimal results or fail. The performers of these studies all deserve full support and credit from the international health care and childcare societies. The results of the study are somewhat difficult to interpret, partially probably by the mature of the study design. Therefore, I have enclosed some questions here below to clarify issues that were unclear.
Questions
Q1. Questionary documents for the interviews (paragraph 2.8, Lane 189-193), how were they selected and tested to be suitable for the study population(s)?
Q2. When considering that Cases and Controls from both DRC and Angola were included. Were there any language issues that could explain some of the differences between ZD och HD categories of children from the two countries studied?
Q3. Data collection (paragraph 2.5, line 135-149). When, during the day (or evening) were the interviews performed? How many different interviewing staff members was involved in the collection of answers?
Q4. In Table 1,(lane 204-206) the family heads background in Age issues and educational school years show were studied. A significantly higher proportion of “Never went to school” family heads was seen among the Cases individuals.
How could this difference in basic education influence the understanding of vaccination importance? Could the authors discuss the impact of no school education and the risk of ZD?
Q5. Period of study, and numbers of participants is quite low (considering the large numbers of inhabitants in the study regions). How significant or how representative are the answers that were obtained?
Q6. Results section look more like a Discussion section content? Anecdotal comments by family members?
Q7. What is the influence of religion among Angolian participants regarding ZD?
Comments on the Quality of English Language
Some sentences seem to lack words. Otherwise quite easy to red the text.
Author Response
"Please see the attachment." in the box if you only upload an attachment

Reviewer 2 Report
Comments and Suggestions for Authors
In this paper, the authors have analyzed the factors which can explain the absence of vaccination in newborns, in the Democratic Republic of Congo. They studied a cohort of 89 babies without correct vaccination, in comparison with a control group constituted of 356 babies with full vaccination.
Major comments:
It is an interesting study, but the results could be described in a more concise form. For example, there are some redondanciesbetween tables 1 & 2, I suggest that table 1 could be included only in supplementary section.
In the same way in paragraph III.2, they report sentences said during the parents' interviews, these sentences could be reported in the supplementary material.
Minor comments:
- page 6, last line, "Approximately" with a capital "A".
- There is certainly a mistake in the unit for "upper arm circonference".
Author Response
Please see the attachment." in the box if you only upload an attachment

Round 2
Reviewer 1 Report
Comments and Suggestions for Authors
Dear Editor-in-Chief,
I have now read the revised manuscript and the authors have clarified most of my questions and comments. However, still the manuscript contain several confusing or unclear sentences and text paragraphs. I here below point on some of the lines where I think that the authors still could clarify their sentences. The manuscript would benefit from an English language editing control.
Comments.
C1. Line 70-71: odd description of distances? Need to be better clarified.
C2. Line: 87 to 92. Poor English, need to be corrected.
C3. Line 110-111: This sentence is unclear and should be better and clearer presented to be understandable.
C4. In paragraph 2.7. Data analysis, the suggested Power analysis procedure suggested in the review reply should be referred to in the text (as reference number 13).
C5. Sometimes vaccine-preventable diseases are presented as VPD and occasionally as VDP. This should be the same everywhere in the text to avoid confusion.
Comments on the Quality of English Language
Dear Editor-in-Chief,
I have now read the revised manuscript and the authors have clarified most of my questions and comments. However, still the manuscript contain several confusing or unclear sentences and text paragraphs. I here below point on some of the lines where I think that the authors still could clarify their sentences. The manuscript would benefit from an English language editing control.
Comments.
C1. Line 70-71: odd description of distances? Need to be better clarified.
C2. Line: 87 to 92. Poor English, need to be corrected.
C3. Line 110-111: This sentence is unclear and should be better and clearer presented to be understandable.
C4. In paragraph 2.7. Data analysis, the suggested Power analysis procedure suggested in the review reply should be referred to in the text (as reference number 13).
C5. Sometimes vaccine-preventable diseases are presented as VPD and occasionally as VDP. This should be the same everywhere in the text to avoid confusion.
Author Response
Dear reviewer,
We thank you for all your valuable comments which helped us to improve the quality of our research article. We thoroughly reviewed our manuscript point by point as suggested. Our answers can be found in bold under each comment.
C1. Line 70-71: odd description of distances? Need to be better clarified.
R: We have clarified the description of distances as follows (see lines 72-74).
The HD is 285 kilometers (km) from Kananga, the capital city of the province, with an estimated population of 349,793 inhabitants over an area of approximately 4,320 km², yielding a density of 81/km².
C2. Line: 87 to 92. Poor English, need to be corrected.
R: We have corrected the English language (lines 86-92)
We considered the following controls:
- Any child aged 12-23 months living in the study HAs who received at least one dose of pentavalent before their first birthday but who did not complete the vaccination schedule recommended by the National Expanded Program of Immunization (EPI).
- Any child aged 12-23 months fully vaccinated before the age of 12 months according to the EPI recommendation and living in the study HAs at the time of data collection
C3. Line 110-111: This sentence is unclear and should be better and clearer presented to be understandable.
R: We have clarified the sentence (lines 110, 101)
We numbered all ZD children from 1 to Ni, with Ni representing the total number of ZD children in the ith CAC.
C4. In paragraph 2.7. Data analysis, the suggested Power analysis procedure suggested in the review reply should be referred to in the text (as reference number 13).
R: Thank you for your comment. We have included the power analysis under the sampling section (lines 96-101).
C5. Sometimes vaccine-preventable diseases are presented as VPD and occasionally as VDP. This should be the same everywhere in the text to avoid confusion.
R: Thank you for your observation. We have now used consistently VPD where needed in the text.
